# The Balance between Energy Efficiency and Renewable Energy for District Renovations in Denmark

Jørgen Rose [1],*, Kirsten Engelund Thomsen [1] and Ole Balslev-Olesen [2]

1 Department of the Built Environment, Aalborg University, A C Meyers Vænge 15, DK-2450 Copenhagen, Denmark
2 Kuben Management A/S, 2. Sal, Ellebjergvej 52, DK-2450 Copenhagen, Denmark
* Correspondence: jro@build.aau.dk

**Abstract:** Emissions can be reduced by increasing the energy efficiency of buildings and supply systems or increasing the share of renewable energy in the energy system. Denmark has a long tradition for district heating and today it supplies more than 65 of dwellings, which indicates that a major part of the transition from fossil fuels to renewables can be achieved at the supply system level rather than on the individual building level. This paper presents calculations performed on a generic Danish district undergoing major renovation. The generic district is based on an existing district in Aalborg and consists of 1019 dwellings spread over three different building typologies. The purpose of the investigation was to determine which combination of energy saving measures would achieve the optimal level of energy efficiency. Calculations were made with average data for district heating in Denmark, district heating based on natural gas, and district heating from renewable energy sources, such as solar heating, biofuels, and heat pumps, respectively. The calculations include costs for investment, maintenance, and operating as a function of the primary energy needs. Global warming potential (GWP) was calculated and included $CO_2$ emissions from space heating, domestic hot water, and electricity for operation and household. The calculations show that for the generic Danish district, which is already connected to a district heating network, the optimal solution is to add 200 mm insulation to roofs (in total 300 mm) and 150 mm insulation to walls (in total 200 mm) and replacing the existing windows with new three-layer low energy windows. Furthermore, the calculations show that in a future scenario with a significantly higher level of renewable energy in the energy system, a shift to individual heat pumps can reduce total emissions by up to 1.5 kg $CO_2$ eq/m$^2$ per year (20 reduction) at an additional cost of EUR 8.0/m$^2$ per year (40 increase). The calculations described in this paper are part of a larger investigation carried out in IEA EBC Annex 75.

**Keywords:** balancing energy efficiency and renewable energy; district optimization; positive energy districts; GWP; energy costs

## 1. Introduction

In 2007, the Intergovernmental Panel on Climate Change synthesis report identified that the sector with the main economic mitigation potentials using technologies and practices expected to be available in 2030 (estimated from bottom-up studies) was the building sector [1]. Buildings account for approximately 40 of all energy use in Europe [2] and 30 of greenhouse gas (GHG) emissions are related to the building sector [3]; therefore, reducing building energy use is key in mitigating climate change. In this context, European Union (EU) adopted the so-called "2020 Climate and Energy Package" [4] in 2007 and the roadmap was updated in October 2014 with the definition of the "2030 Climate & Energy Framework" [5].

In 2011, the Danish government published a strategy with an aim for Denmark to be fossil-free by 2050 [6]. In 2019, the present government set a new and ambitious intermediate target for national $CO_2$ emissions; by 2030, Denmark needs to reduce emissions by 70 in relation to a 1990 baseline [7]. Preliminary calculations for the Danish building stock show

that to reach the overarching goal of a fossil-free society in 2050 it is necessary to reduce the energy use of the existing building stock by up to 50 on average [8].

However, achieving significant reductions in energy use and associated emissions in a cost-effective way is challenging for the existing building stock, especially due to the many architectural and technical hurdles and restrictions. One of the major challenges lies in balancing energy efficiency and renewable energy sources.

Comprehensive research has already been conducted on energy efficiency of existing buildings and balancing energy efficiency with renewable energy production. However, most research has focused on single buildings as in [9], where generic calculations were carried out to investigate the balance, synergies, and trade-offs between renewable energy measures on the one hand, and energy efficiency measures on the other hand. Results of their investigation show that in many cases, the cost-optimal renovation package for energy efficiency measures on the building envelope is the same regardless of the type of energy carrier being used.

Due to the increasing complexity of the energy infrastructure regarding generation, distribution, and use, the single building perspective can lead to sub-optimization. This is demonstrated in, e.g., [10], where authors argue that buildings need to be considered as active participants in a complex and wider district-level energy landscape. To achieve this, the authors suggest the need for a new generation of energy control systems capable of adapting to near real-time environmental conditions, while maximizing the use of renewables and minimizing energy demand within a district environment.

Another example is [11], where the aim of the research was to investigate techno-economic effects and environmental impacts of the energy renovation of residential building clusters on a district heating system. A stock of 343 multi-story apartment buildings located in two Swedish municipalities was included and studied by different cluster combinations of slab and tower blocks. Their study reveals the benefit of integrating simulation and optimization tools to investigate, with a high level of detail, the effect from building cluster energy renovation on the surrounding district heating system.

A paper from Annex 75 [12] aimed at clarifying the cost-effectiveness of various approaches combining energy efficiency and renewable energy sources implementation and focusing on the optimal combination, with respect to the starting situation in a specific city district. Another publication from the same project [13], presents an analysis and comparison of nine district renovation case studies. The study showed that not only energy performances and targets are meaningful for driving these interventions, but other factors can be significant in the upscaling of interventions targeting energy improvements, such as the reduction on $CO_2$ emissions, the improvement of comfort conditions for inhabitants, and the increase in the economic value of buildings.

Positive energy districts (PEDs) have recently become an important concept for urban development. A major driver for the research on PEDs is the climate and energy policy of the European Union and its member states. With the publication of the Set Plan Action 3.2, [14] several European initiatives have started working on the topic of PEDs with the objective to support the development and implementation of at least 100 PEDs by 2025. These include the IEA Annex 83 "Positive Energy Districts", that brings together researchers from European and non-European countries working on positive energy districts [15]. The Annex is a research and dissemination network established for 4 years under the umbrella of the International Energy Agency (IEA). It is open for research organizations and universities and from all IEA member states working on PEDs. The objective of the Annex is to address the PED multidisciplinary dimensions, facilitating the development of PEDs in different worldwide urban contexts.

The main purpose of this paper is to achieve a better understanding of how to balance energy efficiency and renewable energy under different circumstances in a typical Danish district. In Denmark, a long tradition for district heating exists, which indicates that a major part of the transition from fossil fuels to renewables can be achieved at the supply system level. Therefore, the primary focus is on determining which level of energy efficiency is

needed on the building level, in order that the district heating system can deliver fossil fuel free heat and domestic hot water throughout the year.

The paper presents calculations performed on a generic Danish district that is undergoing a major renovation. The generic district is based on an existing district in Aalborg consisting of 1019 dwellings spread over three different building typologies: Single family houses, detached houses, and multi-story apartment buildings. The calculations are part of a larger investigation carried out in IEA EBC Annex 75, where eight different European countries carry out a similar analysis. The purpose of this joint effort is to determine differences, similarities, and generally achieving a better understanding of how to balance energy efficiency and renewable energy under different circumstances.

The rest of the paper is organized as follows: Section 2 presents a description of the research methodology and calculation methods, the energy efficiency measures, and renewable sources. Section 3 provides a detailed description of the investigated district and input data. Section 4 carries out the results and discussion, and the conclusion and lessons learned are provided in Section 5.

## 2. Methods

The following sections explain the approach used to perform calculations of the total final energy consumption for the district under different circumstances. Overall, the calculations follow the methodology developed in IEA EBC Annex 75 documented in [16].

The main advantage of this method is that it provides a fast and robust way of comparing different district solution performances in relation to overall costs, primary energy use, and $CO_2$ emissions. With relatively few inputs, the methodology allows the user to pinpoint the cost optimal set of renovation measures. In addition, by performing simple parametric calculations, a sensitivity analysis can be performed to strengthen the results further. The methodology is further described in [12].

### 2.1. Calculation Method and Data

The Danish energy calculations were performed using the ASCOT calculation tool [17], i.e., a steady-state monthly calculation tool originally developed for energy and financial optimization of the renovation of school buildings following the EN ISO 13790 [18] standard. The tool calculates the final energy consumption for heating and domestic hot water and electricity for operation and household appliances for the entire district, including system losses in both buildings and distribution networks. The primary energy consumption and related $CO_2$ emissions are determined by multiplying the projected primary energy by the $CO_2$ emission factors, which are developed by the Danish Housing and Planning Authority [19], as shown in Table 1. Moreover, the table shows projected energy costs (consumer costs including taxes and charges).

**Table 1.** Primary energy and emission factors for Denmark for 2025 and 2035 [19].

| Danish | Energy Data 2025 | | | Energy Data 2035 | | |
|---|---|---|---|---|---|---|
| Supply System | Emissions [kg $CO_2$-Equiv.] | Primary Energy [kWh/kWh] | Price [€/MWh] | Emissions [kg $CO_2$-Equiv.] | Primary Energy [kWh/kWh] | Price [€/MWh] |
| Heat | | | | | | |
| District heating | 0.088 | 1.156 | 96 | 0.069 | 1.083 | 119 |
| Natural gas | 0.251 | 1.160 | 120 | 0.251 | 1.160 | 149 |
| Oil | 0.331 | 1.280 | 204 | 0.331 | 1.280 | 255 |
| Solar thermal | 0.038 | 0.108 | | 0.038 | 0.108 | |
| Wood pellets | 0.042 | 0.044 | 74 | 0.042 | 0.044 | 92 |
| Electricity | | | | | | |
| Mix Denmark | 0.135 | 2.153 | 326 | 0.041 | 1.695 | 407 |
| Wind turbine | 0.007 | 0.026 | | 0.007 | 0.026 | |
| PV | 0.050 | 0.100 | | 0.050 | 0.100 | |

The financial calculations in ASCOT are performed using net present value (NPV), i.e., determining the total gain of the investment when all costs and revenues over the lifetime are considered. The method is explained in detail in [20]. The financial data used in the calculations are presented in Table 2.

**Table 2.** Financial data used in the calculations [17].

| Parameter | Symbol | Value |
|---|---|---|
| Discount rate | $r_n$ | 4.0 |
| Tax of interest income | $s$ | 0.0 |
| Inflation of energy prices | $i_e$ | 2.5 |
| Inflation of maintenance costs | $i_m$ | 1.5 |
| Expected economic lifetime | $n$ | 50 years |
| Real discount rate, savings | $r_{rs}$ | 1.5 |
| Real discount rate, expenses | $r_{re}$ | 2.5 |
| Net present value factor, savings | $f_{npvs}$ | 35.3 |
| Net present value factor, expenses | $f_{npve}$ | 28.6 |

The discount rate is usually between 4 and 5 and here we have taken a conservative choice of 4.0. The tax of interest income was set to 0 according to [20]. The inflation of energy prices and maintenance costs were taken from [21] based on average values for the period from 2010–2020 and the expected economic lifetime (calculation period) was chosen as the standard 50 years, indicating that energy saving measures with a shorter expected lifetime will be replaced during the period.

The real discount rate for savings is calculated by:

$$r_{rs} = \frac{r_n(1 - s) - i_e}{1 + i_e} = \frac{4.0\,(1 - 0.0) - 2.5}{1 + 2.5} = 1.46$$

Similarly, the real discount rate for expenses is calculated by:

$$r_{re} = \frac{r_n\,(1 - s) - i_m}{1 + i_m} = \frac{4.0\,(1 - 0.0) - 1.5}{1 + 1.5} = 2.46$$

The net present value factors for savings and expenses are calculated as:

$$f_{npvs} = \frac{1 - (1 + r_{rs})^{-n}}{r_{rs}} = \frac{1 - (1 + 1.5)^{-50}}{1.5} = 35.3$$

$$f_{npve} = \frac{1 - (1 + r_{re})^{-n}}{r_{re}} = \frac{1 - (1 + 2.5)^{-50}}{2.5} = 28.6$$

### 2.2. Energy Efficiency Measures and Renewable Energy Sources

The first step of the methodology is to define a gross list of relevant energy efficiency measures for each of the buildings in the district. These measures will typically include building envelope components (replacing windows or adding insulation to walls, floors, roofs), shading/lighting systems, ventilation systems, heating/cooling systems, and local renewable energy systems. Several solutions are evaluated for each component by parametric analysis, e.g., different insulation levels for exterior walls, etc. to determine the best and most cost-effective individual solutions. Measures are evaluated individually, i.e., determining the cost and energy saving potential of each measure by itself and determining the simple payback time, and this comprises the gross list of measures. Finally, the individual measures can be ranked based on the present value, i.e., highest present value should be implemented first, etc.

Depending, of course, on the starting point, the order of measures would typically be (1) measures on the building envelopes, (2) measures for HVAC systems, and (3) measures

related to renewable energy production. It is particularly important to include measures on the building envelope before measures for HVAC systems, since increasing the energy efficiency of the building can influence the choice of HVAC systems. Measures related to renewable energy production are often not influenced by building envelope and HVAC systems; therefore, these can usually be regarded as more or less independent measures.

When the total list has been compiled, measures are combined into so-called renovation packages where individual measures complement each other. The calculations are performed step-by-step in a cumulative way by adding the measure with the shortest payback time first, then the second, etc. This indicates that a graph of the correlation between costs and primary energy or $CO_2$ emissions will have a very specific and easily interpretable course.

It should be noted that in practice there may be other factors which determine the order of measures, e.g., national or local building code requirements, or improvements to the buildings needed for them to function adequately, or the ability to supply an acceptable indoor climate for residents/users. Moreover, it should be noted that future developments in the outside energy system may influence decisions, e.g., a district heating or cooling network may be updated to reduce system losses, which can then shift the cost-optimum energy efficiency levels.

### 2.3. Combining the Community Demand, Renewable Solutions, and Outside Energy System

The district intervention should be planned in a way that it synergizes and integrates well with the outside energy system, i.e., avoiding being an isolated energy island. By planning it in order that the district can interact smartly with the greater energy system, it can be a very useful and valuable part in the grander scheme, i.e., importing and exporting energy and utilizing storage facilities optimally. This will help in reducing consumption and production peaks of the district and thereby avoid sub-optimization of individual system components.

The outside supply system is illustrated in Figure 1.

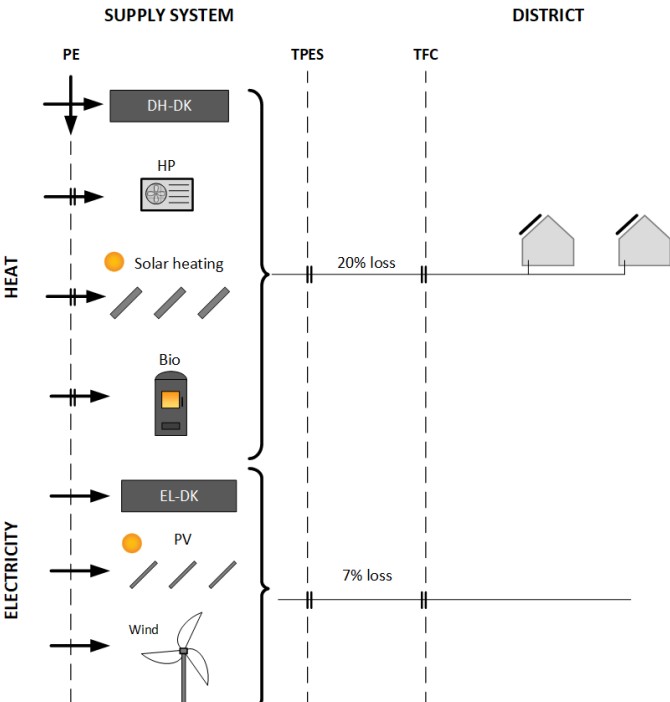

**Figure 1.** Principle of the central heating and electricity supply system in the area. PE: Primary energy is the total energy need for the supply system; TPES: Total primary energy supply; TFC: Total final consumption of the district.

Total final consumption (TFC) is the energy needs in the district calculated with the energy calculation tool and includes space heating, domestic hot water, electricity for operation and household, and losses from the internal district heating network. A system heat loss of 20 of district heating production and 7 of electricity production (TPES) is included in the calculations, corresponding to typical Danish values. These values were developed by the Danish Energy Agency [22]. The analysis includes heat production from district heating with average data for district heating in Denmark (DH-DK) and from renewable energy, such as solar heating, biofuel (Bio), and heat pump (HP). Electricity consumption is covered by the general grid with average data of electricity in Denmark (EL-DK) and from local renewable energy, such as photovoltaic panels (PV) and wind turbines (wind).

## 3. Description of District and Input Data

### 3.1. Description of the District

Kildeparken is a built-up area from the 1970s in Eastern Aalborg that consists of three clusters of buildings: Blåkildevej, Fyrkildevej, and Ravnkildevej. Kildeparken has a total area of approx. 540,000 m$^2$ and around 2450 people live there. Kildeparken consists of 155 single-family houses, 18 apartment blocks containing 432 apartments, and 432 detached houses, and the total heated area is approx. 100,000 m$^2$ (see Figures 2 and 3).

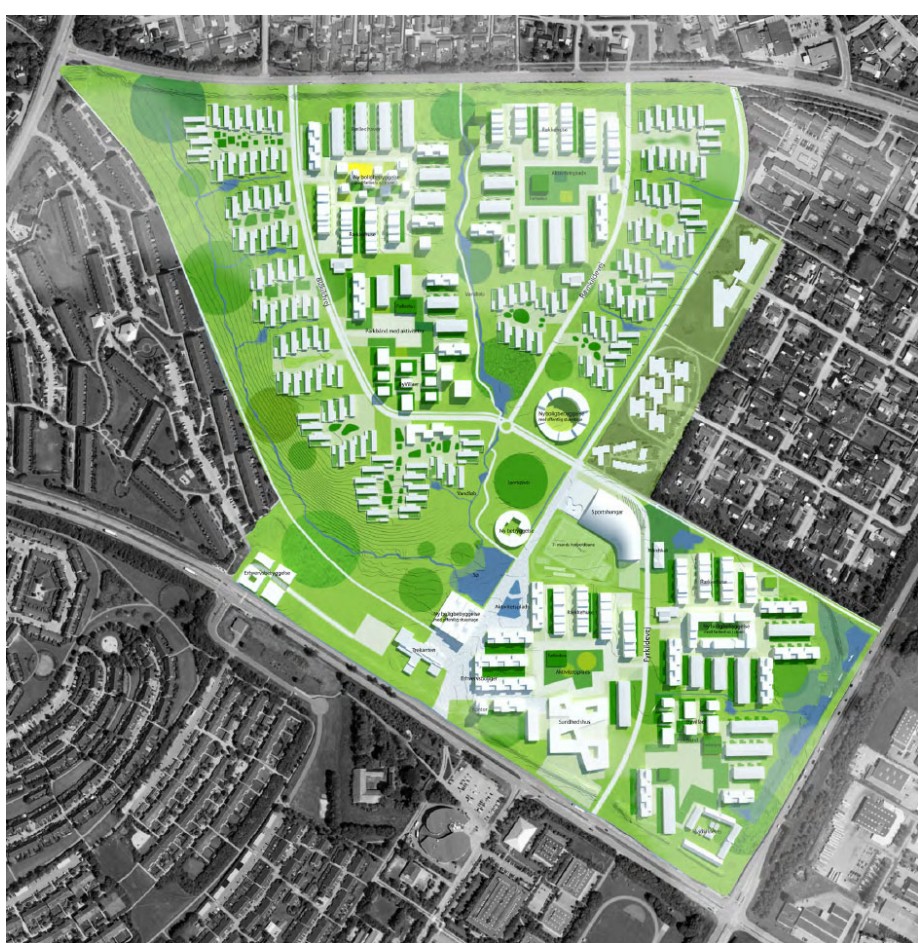

**Figure 2.** Schematic/aerial view of Kildeparken (Source: www.kildeparken2020.dk, accessed on 17 July 2022).

Kildeparken underwent an ambitious renovation which was initiated in 2011 and ended in 2020, but the district is used here for performing generic calculations on possible alternative solutions to the ones chosen in the original project. The original renovation project has been documented in Annex 75 as a so-called Success Story [23] and along with eight other projects it was evaluated by Rose et al. [13].

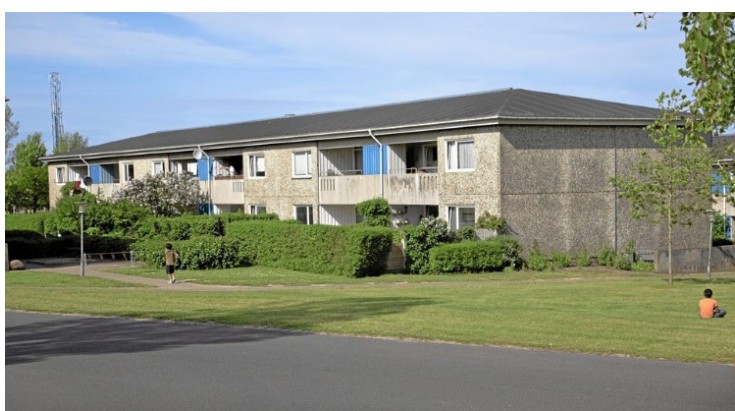 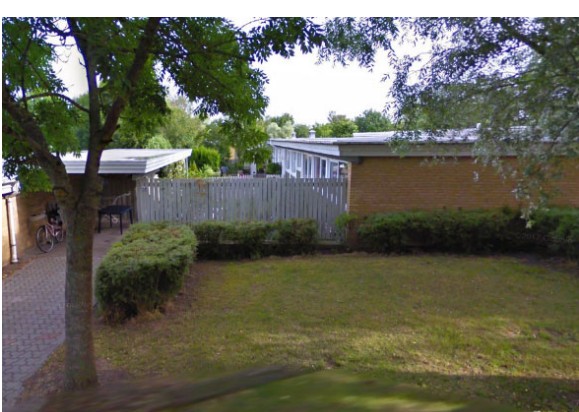

**Figure 3.** Apartment block (**left**) and detached house (**right**) before renovation.

The average electricity consumption per year and m$^2$ excluding heating, cooling, and ventilation is assumed to be approx. 30 kWh/m$^2$/year, corresponding to the mean electricity consumption for households in Denmark. The average indoor temperature is assumed to be 20 °C. The ventilation rate in all building typologies is 0.34 L/s per m$^2$, corresponding to the minimum requirement in Danish Building Regulations [24] and the domestic hot water consumption is 250 L/m$^2$ per year, corresponding to the average consumption in Danish households.

Table 3 summarizes the characteristic input for the three different building typologies. On average, it is assumed that windows are distributed evenly between north and south facades.

**Table 3.** Input parameters used in the calculations.

|  | Single-Family | Apartment Blocks | Detached Houses |
|---|---|---|---|
| Number of buildings [-] | 155 | 18 | 27 |
| Number of dwellings [-] | 155 | 432 | 432 |
| Gross floor heated area per unit, mean [m$^2$] | 120 | 90 | 100 |
| Façade area incl. window area [m$^2$] | 164.3 | 57.3 | 67.5 |
| Area of windows to North per units [m$^2$] | 13.2 | 9.9 | 11.0 |
| Area of windows to East per units [m$^2$] | 0 | 0 | 0 |
| Area of windows to South per units [m$^2$] | 13.2 | 9.9 | 11.0 |
| Area of windows to West per units [m$^2$] | 0 | 0 | 0 |
| Number of floors [-] | 1 | 3 | 2 |

### 3.2. Building Envelopes

The building envelopes are generally worn down and there are quite significant thermal bridges. Apartment blocks have concrete walls with approx. 50 mm insulation and single-family houses and detached houses have brick walls with approx. 50 mm insulation. Roofs have approx. 100 mm insulation and windows have a two-layer glazing and wooden frame. Table 4 lists the approximate mean U-values before renovation. The U-values reflect

the typical constructions of the time; however, compared to Danish standards of today U-values are quite high.

**Table 4.** U-values before renovation.

|  | U-Value [W/m$^2$ K] |
| --- | --- |
| Slab on ground (wood/tiles) | 0.40 |
| Slab above basement (wood/tiles) | 0.60 |
| Wall | 0.70 |
| Roof | 0.40 |
| Windows | 2.90 |

### 3.3. Building Systems

The buildings all have combinations of mechanical exhaust ventilation and natural ventilation. Air is extracted from kitchens (through the cooker hood) and bathrooms and supplied by natural ventilation/infiltration through open windows/doors and leaks in the building envelope.

The district is supplied with heat from the outside through an existing district heating network. Electricity is supplied via the common electricity grid in Denmark.

### 3.4. Renovation Measures and Local Renewable Energy Sources

The renovation measures that can be implemented to increase the overall energy efficiency of the district cover replacing the existing windows, adding insulation to the roof, adding insulation to the exterior walls, installing individual heat pumps, installing solar thermal panels, installing photovoltaic panels, and installing balanced mechanical ventilation with heat recovery. Table 5 shows the list and data for individual measures (M1–M7) and Table 6 shows the list of renovation packages (M8–M11 and A0–A5). Scenarios M8–M11 are all based on the existing district heating system, and in scenarios A0–A5, district heating is replaced by individual heat pumps and complemented by a solar thermal installation for domestic hot water production. Scenario A0 corresponds to the reference case where the only change is that heating is now based on individual heat pumps. In the tables, BR61 represents the building regulations from 1961 [25] which were in effect when Kildeparken was built and corresponds to the reference scenario. "HP" is heat pump, "SH" is solar thermal, "PV" is photovoltaics, and "MVHR" is mechanical ventilation with heat recovery.

### 3.5. Global Renewable Energy Sources

In the initial scenario, corresponding to the year 2025, the data from Table 1 are used in the calculations. However, to further investigate the effect of increasing the overall coverage of renewable energy sources in the outside energy system, two further scenarios are evaluated.

In the first scenario, it is assumed that approx. 50 of the Danish energy supply is based on renewable energy sources. For heat, 10 is covered by solar thermal installations and 40 is covered by heat pumps and these renewable energy sources are connected to the district heating network. For electricity, 10 is supplied by PV installations and 40 is supplied by wind turbines. The relevant financial data for this scenario are given in Table 7.

To investigate the possibilities and economic circumstances related to introducing even higher levels of renewable energy in the global energy system, a scenario is investigated where the data for solar heating and photovoltaics are the same as shown in Table 7, but the heat pump supply is increased to supply 90 of heat and wind turbines are increased to supply 90 of electricity, i.e., corresponding to a situation with 100 renewable energy production.

**Table 5.** Renovation measures analyzed in the calculations. Measures M1–M7.

| | | | Reference | M1 | M2 | M2a | M2b | M2c | M3 | M4 | M5 | M6 | M7 |
|---|---|---|---|---|---|---|---|---|---|---|---|---|---|
| | | Unit | BR61 | 3-Layer Low Energy | 200 mm Roof | 100 mm Roof | 150 mm Roof | 250 mm Roof | 150 mm Wall | HP | Solar | PV | MVHR |
| Wall | U-value | W/m²K | 0.7 | | | | | | 0.19 | | | | |
| | Extra insulation | mm | | | | | | | 150 | | | | |
| | λ-value | W/mK | | | | | | | 0.040 | | | | |
| | Maintenance | of inv. | | | | | | | 0 | | | | |
| | Lifetime | years | | | | | | | 80 | | | | |
| Windows | U-value | W/m²K | 2.9 | 0.7 | | | | | | | | | |
| | g-value | - | 0.75 | 0.53 | | | | | | | | | |
| | Maintenance | of inv. | 2 | 1 | | | | | | | | | |
| | Lifetime | Years | 20 | 60 | | | | | | | | | |
| Roof | U-value | W/m²K | 0.40 | | 0.13 | 0.20 | 0.16 | 0.11 | | | | | |
| | Extra insulation | Mm | | | 200 | 100 | 150 | 250 | | | | | |
| | λ-value | W/mK | | | 0.040 | 0.040 | 0.040 | 0.040 | | | | | |
| | Maintenance | of inv. | | | 0 | 0 | 0 | 0 | | | | | |
| | Lifetime | Years | | | 80 | 80 | 80 | 80 | | | | | |
| HP | COP | - | | | | | | | | 3.5 | | | |
| | Flow temp | °C | | | | | | | | 70 | | | |
| | Return temp | °C | | | | | | | | 50 | | | |
| | Maintenance | of inv. | | | | | | | | 5 | | | |
| | Lifetime | years | | | | | | | | 20 | | | |
| PV | PV Type | | | | | | | | | | | Mono | |
| | Peak power | W | | | | | | | | | | 150 | |
| | Efficiency | | | | | | | | | | | 85 | |
| | PV-area per dwelling | m² | | | | | | | | | | 7.5/20 | |
| | Maintenance | of inv. | | | | | | | | | | 1 | |
| | Lifetime | Years | | | | | | | | | | 25 | |
| SH | SH-area per dwelling | m² | | | | | | | | | 2 | | |
| | Maintenance | of inv. | | | | | | | | | 2 | | |
| | Lifetime | Years | | | | | | | | | 20 | | |
| Ventilation | Air change | l/s per m² | 0.34 | | | | | | | | | | 0.34 |
| | Heat recovery | | | | | | | | | | | | 90 |
| | Air tightness | 50 Pa | 4.0 | | | | | | | | | | 1.5 |
| | SFP | kJ/m³ | | | | | | | | | | | 1.2 |
| | Maintenance | of inv. | | | | | | | | | | | 5 |
| | Lifetime | years | | | | | | | | | | | 20 |
| | Investment costs | €/m² | 0 | 132 | 19 | 9 | 14 | 23 | 106 | 87 | 15 | 32 | 73 |

**Table 6.** Renovation packages analyzed in the calculations. Packages M8–M11 (district heating) and A0–A5 (individual heat pumps).

| | | | Reference | M8 | M9 | M10 | M11 | A0 | A1 | A2 | A3 | A4 | A5 |
|---|---|---|---|---|---|---|---|---|---|---|---|---|---|
| | | Unit | BR61 | M1 + M6 | M1 + M2 + M6 + M3 | M1 + M2 + M6 | M1 + M2 + M6 + M7 + M3 | A0 Ref | M1 | M1 + M6 | M1 + M6 + M2 | M1 + M2 + M3 + M6 | M1 + M2 + M6 + M7 + M3 |
| Wall | U-value | W/m²K | 0.7 | | | 0.19 | 0.19 | | | | | 0.19 | 0.19 |
| | Extra insulation | mm | | | | 150 | 150 | | | | | 150 | 150 |
| | λ-value | W/mK | | | | 0.040 | 0.040 | | | | | 0.040 | 0.040 |
| | Maintenance | of inv. | | | | 0 | 0 | | | | | 0 | 0 |
| | Lifetime | years | | | | 80 | 80 | | | | | 80 | 80 |
| Windows | U-value | W/m²K | 2.9 | 0.7 | 0.7 | 0.7 | 0.7 | | 0.7 | 0.7 | 0.7 | 0.7 | 0.7 |
| | g-value | - | 0.75 | 0.53 | 0.53 | 0.53 | 0.53 | | 0.53 | 0.53 | 0.53 | 0.53 | 0.53 |
| | Maintenance | of inv. | 2 | 1 | 1 | 1 | 1 | | 1 | 1 | 1 | 1 | 1 |
| | Lifetime | Years | 20 | 60 | 60 | 60 | 60 | | 60 | 60 | 60 | 60 | 60 |

**Table 6.** *Cont.*

| | | | Reference | M8 | M9 | M10 | M11 | A0 | A1 | A2 | A3 | A4 | A5 |
|---|---|---|---|---|---|---|---|---|---|---|---|---|---|
| Roof | U-value | W/m²K | 0.40 | | 0.13 | 0.13 | 0.13 | | | | 0.13 | 0.13 | 0.13 |
| | Extra insulation | mm | | | 200 | 200 | 200 | | | | 200 | 200 | 200 |
| | λ-value | W/mK | | | 0.040 | 0.040 | 0.040 | | | | 0.040 | 0.040 | 0.040 |
| | Maintenance | of inv. | | | 0 | 0 | 0 | | | | 0 | 0 | 0 |
| | Lifetime | Years | | | 80 | 80 | 80 | | | | 80 | 80 | 80 |
| HP | COP | - | | | | | | 3.5 | 3.5 | 3.5 | 3.5 | 3.5 | 3.5 |
| | Flow temp | °C | | | | | | 70 | 70 | 70 | 70 | 70 | 70 |
| | Return temp | °C | | | | | | 50 | 50 | 50 | 50 | 50 | 50 |
| | Maintenance | of inv. | | | | | | 5 | 5 | 5 | 5 | 5 | 5 |
| | Lifetime | years | | | | | | 20 | 20 | 20 | 20 | 20 | 20 |
| PV | PV Type | | | Mono | Mono | Mono | Mono | | | | Mono | Mono | Mono |
| | Peak power | W | | 150 | 150 | 150 | 150 | | | | 150 | 150 | 150 |
| | Efficiency | | | 85 | 85 | 85 | 85 | | | | 85 | 85 | 85 |
| | PV-area per dwelling | m² | | 7.5 | 7.5 | 7.5 | 7.5 | | | | 7.5 | 7.5 | 7.5 |
| | Maintenance | of inv. | | 1 | 1 | 1 | 1 | | | | 1 | 1 | 1 |
| | Lifetime | Years | | 25 | 25 | 25 | 25 | | | | 25 | 25 | 25 |
| SH | SH-area per dwelling | m² | | | | | | 2 | 2 | 2 | 2 | 2 | 2 |
| | Maintenance | of inv. | | | | | | 2 | 2 | 2 | 2 | 2 | 2 |
| | Lifetime | Years | | | | | | 20 | 20 | 20 | 20 | 20 | 20 |
| Ventilation | Air change | 1/s per m² | 0.34 | | | | 0.34 | | | | | | 0.34 |
| | Heat recovery | | | | | | 90 | | | | | | 90 |
| | Air tightness | 50 Pa | 4.0 | | | | 1.5 | | | | | | 1.5 |
| | SFP | kJ/m³ | | | | | 1.2 | | | | | | 1.2 |
| | Maintenance | of inv. | | | | | 5 | | | | | | 5 |
| | Lifetime | years | | | | | 20 | | | | | | 20 |
| | Investment costs | €/m² | 0 | 132 | 19 | 9 | 14 | 23 | 106 | 87 | 15 | 32 | 73 |

**Table 7.** Financial and operational data for a scenario where energy is covered by 50 renewable sources.

| Heat—Solar Heating | | | | | |
|---|---|---|---|---|---|
| Solar fraction | 10.0 | | | | |
| Solar performance | 600 | kWh/m² | | Total | |
| Investment cost | 80 | €/m² | | 589,394 | € |
| Maintenance cost | 2.0 | | | 11,788 | €/year |
| Lifetime | 20 | years | | | |
| Solar heat contribution | 1,657,671 | kWh/year | | | |

| Heat–Heat pumps (Air) | | | | | |
|---|---|---|---|---|---|
| HP fraction | 40.0 | | | | |
| Size–heat | 0.84 | MW | | | |
| COP | 3.58 | | | | |
| Operation time | 4800 | Hours per year | | Total | |
| Investment cost | 826,667 | €/MW | | 1,359,466 | € |
| Maintenance cost | 2 | €/MWh | | 13,261 | €/year |
| Lifetime | 20 | years | | | |
| HP contribution | 6,630,686 | | kWh/year | | |

| Electricity–Photovoltaics | | | | | |
|---|---|---|---|---|---|
| PV fraction | 10.0 | | | Total | |
| Investment cost | 827 | €/kWp | | 298,865 | € |
| Maintenance cost | 1.0 | | | 2989 | €/year |
| Lifetime | 25 | years | | | |

| Electricity—Wind turbines | | | | | |
|---|---|---|---|---|---|
| Wind fraction | 40.0 | | | | |
| Size | 2.0 | MW | | Total | |
| Investment cost | 1,985,333 | €/MW | | 611,284 | € |
| Maintenance cost | 5.0 | | | 30,564 | €/year |
| Lifetime | 15 | Years | | | |
| Contribution | 1,446,119 | kWh/year | | | |

## 4. Results and Discussion

The result for each calculation covers the total final consumption, the primary energy use, the operational costs, the emissions (GWP), and the life cycle costs (NPV). As an example, Table 8 shows the calculation results for the reference case and the M1 scenario, i.e., replacing the windows with three-layer low energy windows.

**Table 8.** Example showing the calculation results for the reference case and the M1 scenario.

| | | Reference Case | M1 |
|---|---|---|---|
| Total final consumption | | | |
| Heat | $kWh/m^2$ per year | 164.9 | 126.5 |
| Electricity | $kWh/m^2$ per year | 36.0 | 35.3 |
| Primary energy use | | | |
| Heat | $kWh/m^2$ per year | 190.6 | 146.2 |
| Electricity | $kWh/m^2$ per year | 77.4 | 75.9 |
| Operational costs | | | |
| Heat | €/$m^2$ per year | 14.3 | 11.0 |
| Electricity | €/$m^2$ per year | 10.6 | 10.4 |
| Total | €/$m^2$ per year | 24.9 | 21.4 |
| Life cycle costs | | | |
| Investment, I | €/$m^2$ per year | 0.0 | 132.4 |
| Operational costs, O | NPV, €/$m^2$ | 878.6 | 754.0 |
| Maintenance, M | NPV, €/$m^2$ | 96.6 | 58.8 |
| Replacement costs, R | NPV, €/$m^2$ | 58.1 | 0.0 |
| NPV (I + O + M + R) | €/$m^2$ per year | 20.7 | 18.9 |
| Emissions (GWP) | | | |
| Heat | kg $CO_2$-Equiv. | 14.5 | 11.1 |
| Electricity | kg $CO_2$-Equiv. | 4.9 | 4.8 |
| Total | kg $CO_2$-Equiv. | 19.3 | 15.9 |

In the following sections the results of four different analysis are presented.

### 4.1. Increasing Renewable Energy in the Energy Mix

The purpose of the first analyses is to determine how an increased share of renewable energy in the energy mix will influence the cost-optimality of renovation measures, i.e., how will the decarbonization of energy systems influence the balance between energy efficiency and renewable energy. The calculation compares the situation corresponding to the existing energy mix of today and the expected future energy mix as described in Table 7. Results are plotted in Figure 4 where M0 and A0 are the reference scenarios, M1 and A1 correspond to replacing the windows, M8 and A2 correspond to adding PV, M10, and A3 to insulating roofs, M9 and A4 to insulating walls, and finally M11 and A5 to installing mechanical ventilation with heat recovery. The order of the measures corresponds to starting with the lowest simple payback time and gives the curves their distinct look with an easily interpretable minimum that signifies the cost-optimal solution. DH (DK) is the present energy mix in Denmark and DH (RE) is an energy mix corresponding to the scenario described in Table 7, i.e., where 50 of energy is based on renewable sources. The results are shown in Figure 4 as the primary energy use as a function of the total costs per year (left) and the emissions as a function of the total costs per year (right).

Comparing the two scenarios both primary energy use and emissions will decrease significantly when the share of renewable energy in the energy mix is increased, and this affects the cost-optimality. With the present energy mix the final renovation stage (installing mechanical ventilation with heat recovery) has lower total costs per year than the reference

case, whereas this is clearly not the case for the future scenario where wall insulation (A4) has the same total costs per year as the reference.

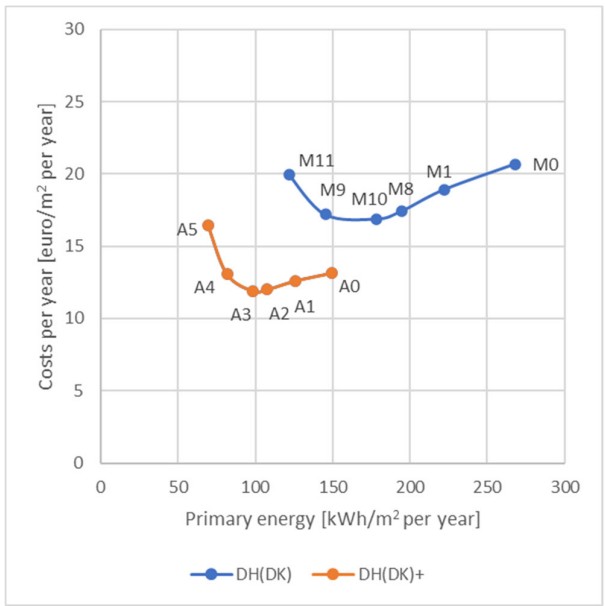 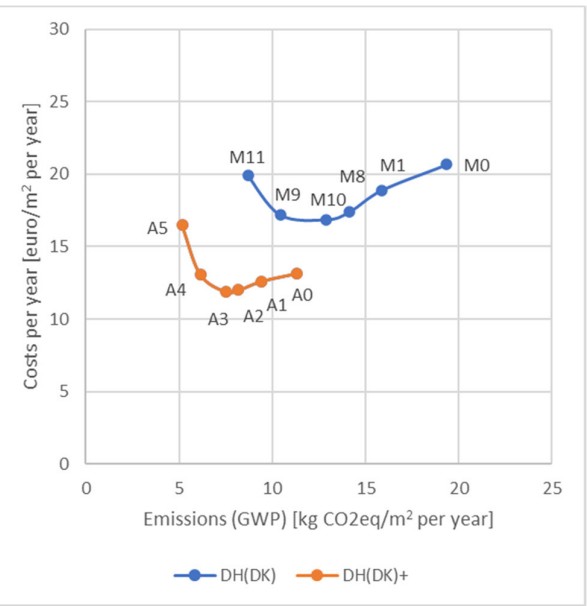

**Figure 4.** Comparing emission factors of today with a future scenario with an increased share of renewable energy in the energy mix.

### 4.2. Building Energy Efficiency Level before Renovation

As mentioned, Kildeparken was erected during the early 70ies; therefore, it fulfilled the building regulations from 1961. In 1977, a new set of regulations were introduced [26] with stricter requirements for building envelope insulation, etc. The purpose of the second analyses is to determine how cost-optimality is influenced if the energy efficiency starting point for the district is higher, i.e., how will this influence the balance between energy efficiency and renewable energy. The results are shown in Figure 5, where DH (DK) corresponds to the actual state of the district (building regulations 1961) and DH (DK)77 corresponds to a situation where the building envelopes have an energy efficiency corresponding to the building regulations from 1977.

Comparing the two scenarios in Figure 5, it is clearly still cost-effective to introduce energy-saving measures in a district that does not meet modern energy-efficiency requirements. However, it should be noted that if the starting point (the reference building) is shifted even further toward today's standards, the system losses (heat loss in the district heating network and electricity network) will make investments in energy efficiency less and less cost-effective.

### 4.3. District Heating vs. Individual Heat Pumps

The purpose of the third analyses is to determine whether it makes sense to consider switching the existing district heating system for individual heat pumps. The rationale behind this thought is that a shift to individual heating will remove the system losses in the district heating network (corresponding to approx. 20 in Denmark). For this purpose, two scenarios are developed: DH (RE) and IH (RE) correspond to future scenarios where the energy system has an increased level of renewable sources (as described in Table 7). DH corresponds to maintaining the existing district heating network and IH corresponds to a switch to individual heat pumps. The results are shown in Figure 6.

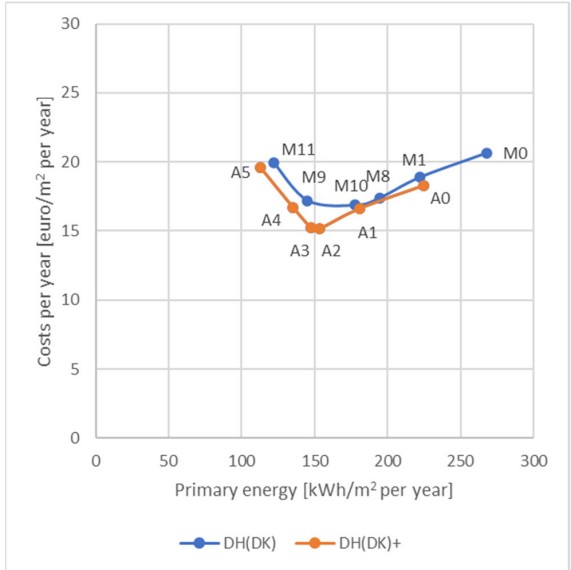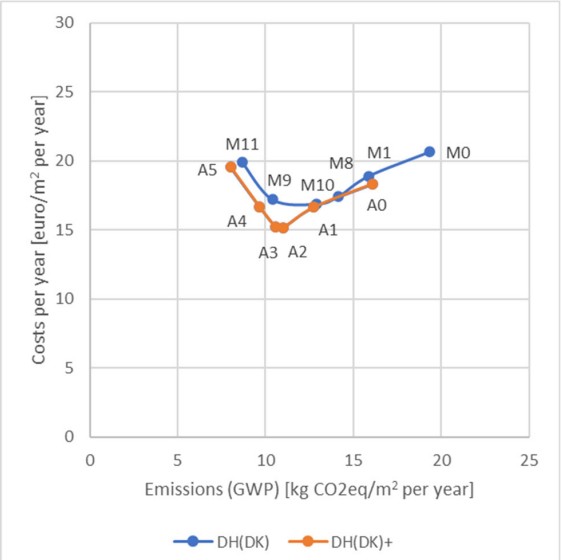

**Figure 5.** Difference in the starting point for the energy efficiency of the district.

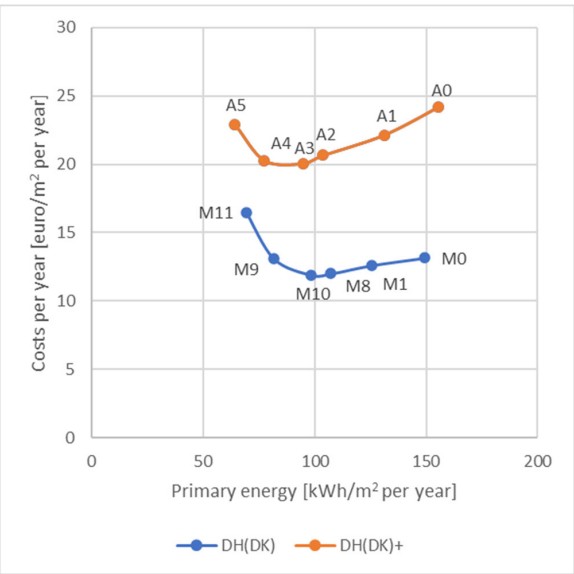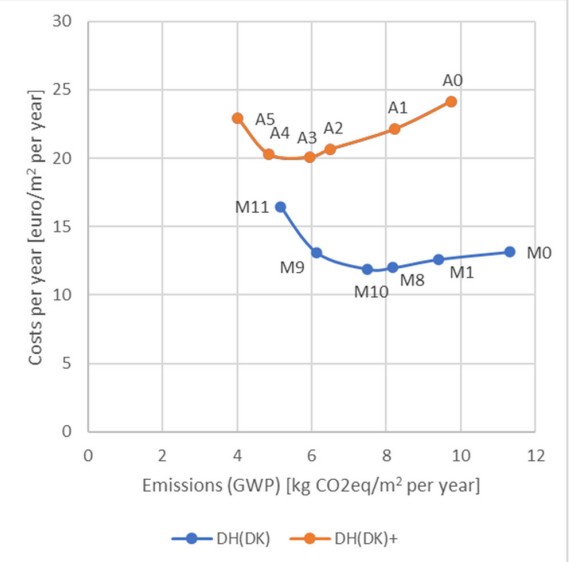

**Figure 6.** Results for a future scenario with an increased share of renewables in the mix. Comparing district heating to individual heat pumps.

In Figure 6, the primary energy use is similar in the two scenarios. Individual heat pumps can achieve somewhat lower emission levels; however, at a significantly higher cost. Cost-optimality is similar but more pronounced in the situation with individual heating. If the district did not have a district heating network to start with, the total costs per year for the district solution would have been similar to the individual heat pumps solution or even higher.

### 4.4. Uncertainties Related to Prices for Energy Measures

The prices of renovation measures are not static and will vary depending on, e.g., the geographical location of the renovation project, the season of the year, and the activity level in the building industry in general. Therefore, an extra analysis was carried out to determine how much it would influence the overall results if the price of energy saving measures were 25 higher. Results are shown in Figure 7, where DH (DK) signifies the prices of renovation measures when the calculations were performed (2022) and DH (DK) + signifies a 25 increase in prices.

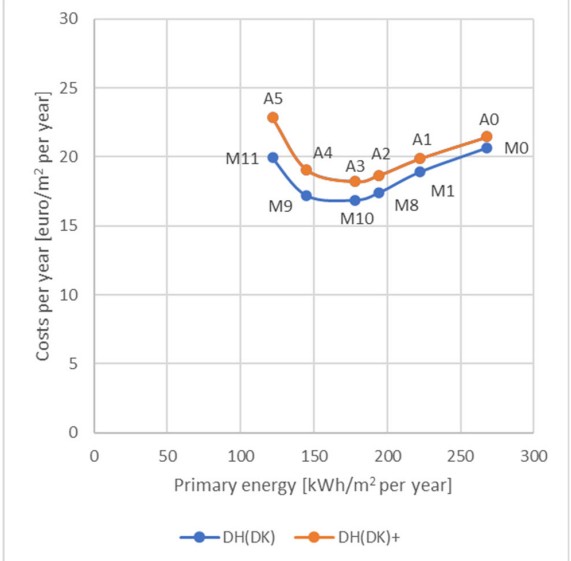 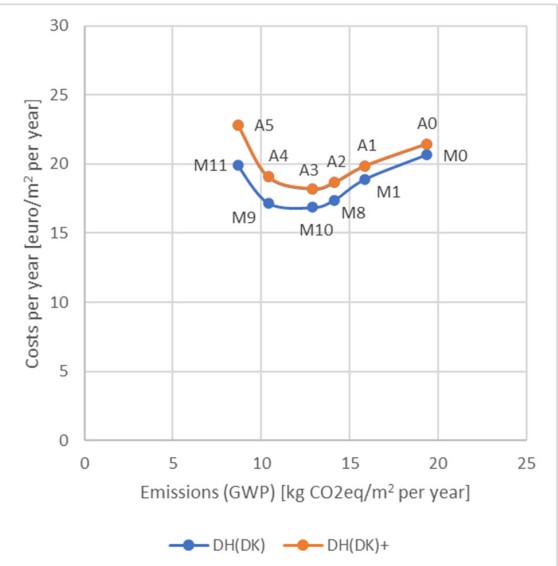

**Figure 7.** Sensitivity analysis on prices of energy renovation measures. The DH(DK) + scenario signifies a 25 price increase.

Comparing the two sets of graphs in Figure 7 shows that a 25 increase in renovation measure prices will only influence results slightly. However, for scenario A5 (installing mechanical ventilation with heat recovery), it is evident that the overall costs per year exceed the overall costs per year for the reference case (A0), i.e., including this measure in the total package of measures will result in increased costs overall.

## 5. Conclusions

The calculations performed in this paper show that for the generic Danish district, which is already connected to a district heating network, the optimal solution is to add 200 mm insulation to roofs (in total 300 mm) and 150 mm insulation to walls (in total 200 mm) and replace the existing windows with new three-layer low energy windows. Total costs before the intervention are EUR $20.2/m^2$ per year and the optimal level costs are EUR $16.4/m^2$ per year.

Moreover, the calculations show that balanced mechanical ventilation with heat recovery is not profitable; however, in typical Danish multi-family buildings this measure would always be carried out anyway, since this will have a significant impact on the indoor climate. The total costs per year including this measure is EUR $20.0/m^2$ per year, i.e., still lower than the costs before the intervention.

Furthermore, the calculations show that in a future scenario with a significantly higher level of renewable energy in the energy system, a shift to individual heat pumps can reduce total emissions by up to 1.5 kg $CO_2$ eq/$m^2$ per year (20 reduction) at an additional cost of EUR $8.0/m^2$ per year (40 increase).

Based on the calculations, the future development related to renewable energy integration in the energy mix will influence cost-optimality for renovation cases. If we only look at emissions, over a relatively short period of time, these will be reduced significantly through the decarbonization of the energy systems and energy efficiency at the building level will make less and less sense. However, if we also look at the energy prices, investment in energy efficiency measures is still very relevant, especially when considering that energy prices are expected to increase by 50 or more in 2022 alone.

In addition, the investigations in this paper show that for a country such as Denmark, where district heating is well established and covers a large proportion of buildings, it is better to utilize and expand these networks rather than converting to individual solutions. In other countries, which may not have existing district heating networks, it may be more advisable to look at individual solutions. However, it is important to note that using central solutions (e.g., district heating) rather than decentral solutions (e.g., individual heat pumps)

has the added benefit of enabling incorporation of, e.g., waste heat in the network, common storage facilities, and similar synergies, all of which are very important factors to consider when designing future energy systems.

Finally, the calculations show that the balance between energy efficiency measures and renewable energy sources is very dependent on the starting situation. If the district has a relatively high level of energy efficiency (buildings erected in the last three decades) to start with, further investments should probably focus on renewable energy rather than energy efficiency. However, this should always be based on an individual case-by-case assessment.

Generally, investigations in this paper were performed prior to the war in Ukraine, which indicates that prices of energy saving measures and energy are even more volatile today than what they were when the calculations were performed. This, of course, adds further to the uncertainty of the results, but as shown in Section 4.4, even relatively large variations in prices do not necessarily significantly alter the conclusions.

Finally, it should be noted that renovation not only improves energy efficiency of buildings, but has a large amount of co-benefits as well, e.g., improvement of indoor climate, lower influence of price increases and thereby risks of increased energy-poverty, etc. These co-benefits are difficult to weigh against emissions and costs, but can sometimes be more important.

The calculations described in this paper relate to a very homogenous district that comprises only single- and multi-family dwellings and has single sources for heat and electricity. Therefore, future research should focus on determining relevant solutions in more heterogenous districts involving, e.g., industry and commercial buildings, etc. and districts where the existing energy infrastructure is less developed or incoherent.

**Author Contributions:** Conceptualization, J.R. and K.E.T.; methodology, O.B.-O.; software, O.B.-O.; validation, J.R.; formal analysis, O.B.-O.; investigation, J.R. and K.E.T.; resources, O.B.-O.; data curation, O.B.-O.; writing—original draft preparation, J.R. and K.E.T.; writing—review and editing, J.R. and K.E.T.; visualization, J.R., K.E.T. and O.B.-O.; project administration, K.E.T.; funding acquisition, J.R. and K.E.T. All authors have read and agreed to the published version of the manuscript.

**Funding:** This research was funded by the Danish Energy Agency grant number 64017-0586.

**Acknowledgments:** This work was financed by the Danish Energy Agency through the EUDP program.

**Conflicts of Interest:** The authors declare no conflict of interest.

### Nomenclature

| Symbol | Description | Unit |
| --- | --- | --- |
| $r_n$ | Discount rate | |
| s | Tax of interest income | |
| $i_e$ | Inflation of energy prices | |
| $i_m$ | Inflation of maintenance costs | |
| n | Expected economic lifetime | Years |
| $r_{rs}$ | Real discount rate, savings | |
| $r_{re}$ | Real discount rate, expenses | |
| $f_{npvs}$ | Net present value factor, savings | |
| $f_{npve}$ | Net present value factor, expenses | |
| U | Thermal transmittance | $W/m^2\,K$ |
| l | Thermal conductivity | $W/mK$ |
| g | Thermal resistance | $m^2\,K/W$ |
| SFP | Specific fan power | $kJ/m^3$ |
| COP | Coefficient of performance | - |
| I | Investment | $€/m^2$ |
| O | Operational costs | $€/m^2$ |
| M | Maintenance | $€/m^2$ |
| R | Replacement costs | $€/m^2$ |

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
