# Peer review of "The Balance between Energy Efficiency and Renewable Energy for District Renovations in Denmark"

_sustainability, doi:10.3390/su142013605_

Round 1

Reviewer 1 Report

Paper No.: sustainability-1893892

Title: "The balance between energy efficiency and renewable energy for district renovations in Denmark"

I admire the author's efforts in preparing this work and thank them for submitting this paper to the Sustainability journal (ISSN 2071-1050). This paper seems to be an interesting concept. Nevertheless, this work requires further modifications as listed below:

1) The new contributions of the paper need to be further clarified. The present form does not have sufficient contributions and results to justify the novelty of a high-quality journal paper. I would like to Major Revision this paper on this basis.

2) The present references are not enough to cover this research area. More latest references are needed.

3) Not even one direction for future research is provided. There should be at least 4-6 solid directions for future research.

4) English should be improved. The authors should carefully and thoroughly check the paper, and correct all the spelling and grammar mistakes in the final version. The use of Grammarly software is also recommended.

5) The advantages of the proposed method of this paper should be more highlighted.

6) More comparison and discussion with the existing similar conclusions should be strengthened in order to show the innovation and contribution of the proposed method.

The final evaluation will proceed after performing the requested revisions.

Yours sincerely,

Reviewer 2 Report

File attached. 

Reviewer 3 Report

This manuscript is within the scope of the journal. The topic is interesting. However, some problems still need to be clarified and revised. I would recommend for a major revision of the manuscript after addressing the significant improvement requirements.

The points of concern are as follows:

1. The introduction is disordered and aimless. Author/ Authors just simply list the lows of literature reviews. However, the important information, such as innovations, focused issues, methods, and value of this article, are missing or oversimplified. It is necessary to rewrite the introduction. The literature review has been how written that leads to misunderstanding.

2. The novelty of the work must be clearly addressed and discussed, compare your research with existing research findings, and highlight novelty, (compare your work with existing research findings and highlight novelty).

3. The main objective of the work must be written more clearly. Also, you should have a subsection on the strengths and limitations of your study.

4. Nomenclature section is also missing. Add completely in the table.

5. There is a strong need to improve the manuscript's language as it is not appropriate for the Journal of Sustainability. The text has lots of grammatical errors. Most of the sections cannot be read well.

6. In lines 149, 157, 201, 222, 223, 249, 265, 266, 280, 288, 294, 301, 313, 314, 321, 323, 341, 347, 362, 366, 377, and 383, what does “Error! Reference source not found” mean?!! Refine them.

7. Author/ Authors need to add more results in the conclusion section and abstract section (especially numerical results) to thoroughly support the main findings. The results are not clear and complete and must be more. The conclusion section should be rewritten with the complete data.

8. Where is your manuscript validation section? Specify it!

If you do not have anything, add it completely to the manuscript and give complete information on the validation procedures.

9. It is also suggested to include a paragraph at the end of the conclusion section that should describe the applications of the present work, recommendations, and future scope.

Round 2

Reviewer 1 Report

Manuscript ID:  sustainability-1893892

Title: The balance between energy efficiency and renewable energy for district renovations in Denmark

I admire the authors’ efforts for the preparation of this work and thank them for submitting this paper to the Journal of Sustainability (ISSN 2071-1050). The authors have improved the manuscript by addressing all concerns raised by the reviewer. I have no further revision requests. The article is fully eligible for publication in the Sustainability journal.

Yours sincerely,

Author Response

As suggested, a nomenclature has been added to the paper.

Reviewer 2 Report

I accept the publication of the paper.

Author Response

(The authors gave the same response as above.)

Reviewer 3 Report

According to the corrections made by the author/ authors, one of the important mentioned that needs to be done has not been done and the author/author's explanation is not convincing and technical in that one case. I would recommend for a minor revision of the manuscript.

The point of concern are as follows:

- The nomenclature section is missing. Add completely in the table.

Author Response

(The authors gave the same response as above.)
